# A Review of Thermal Monitoring Techniques for Radial Permanent Magnet Machines

**Tianze Meng and Pinjia Zhang \***

Department of Electrical Engineering, Tsinghua University, Beijing 100084, China; mtz18@mails.tsinghua.edu.cn
\* Correspondence: pinjia.zhang@ieee.org

**Abstract:** Permanent magnet machines are widely applied in motor drive systems. Therefore, condition monitoring of permanent magnet machines has great significance to assist maintenance. High temperatures are accountable for lots of typical malfunctions and faults, such as demagnetization of the permanent magnet (PM) and inter-turn short circuit of stator windings. Therefore, temperature monitoring of the PM and stator windings is essential for reliable operation. In this paper, an overview introducing and evaluating existing thermal monitoring methods is presented. First, the mechanism of thermal-caused failures for the PM and stator windings is introduced. Then, the design procedure and principles of existing temperature monitoring methods are introduced and summarized. Next, the evaluations and recommendations of application feasibility are demonstrated. Finally, the potential future challenges and opportunities for temperature monitoring of the PM and stator windings are discussed.

**Keywords:** thermal monitoring; permanent magnet machines; PM; stator windings





## 1. Introduction

Permanent magnet synchronous machines (PMSMs) have attracted more and more attention, especially in recent years. The significant research efforts are due to the high dynamic response, high reliability, high torque density, and high efficiency of PMSMs. Meanwhile, PMSMs have been used for numerous applications, such as automotive, wind power generation, electric vehicles, aerospace, and servo-drives. Therefore, reliable operation is critical for protecting the safety of life and production. In practice, the permanent magnet (PM) remanent flux density determines the torque production capability and the insultation of stator windings determines the probability of failure or residual lifetime. Consequently, the performance of PMSMs is mainly dependent on the PM magnetization state and insulation condition of the stator windings. However, these two properties are susceptible to the operating environment, e.g., the component temperature. High temperatures cause the demagnetization of the PM and intensive thermal stress will lead to insulation aging of the stator windings.

In the last few decades, the techniques of temperature monitoring for PMSMs have been comprehensively researched and developed. Whether for the PM or stator windings, from the implementation level, the temperature monitoring methods can be categorized into contact direct measuring methods and non-contact estimation methods.

Generally, contact direct measuring methods directly measure the temperature, and are usually implemented using surface-mounted thermal devices [1–6]. In addition, infrared-detecting methods [7,8] are widely used in some situations for convenience. Because of the installation requirements of temperature sensors, the applicability of these direct measuring methods is determined by the accessibility of mechanical structure and overall cost. Non-contact estimation methods are usually based on the identification of thermal-relevant parameters or the temperature derivation of intelligence algorithm. Relying on the measurable quantities, models [9] or iterative multilevel-algorithms [10] are established to monitor the temperature.

Considering the principles used in these methods, these temperature monitoring methods can be classified and presented from the perspective of technical category. Technically, the existing methods can also be categorized into sensor-based methods, model-based methods, and AI algorithm-based methods. In Figure 1, methods classified by monitoring techniques are presented respectively.

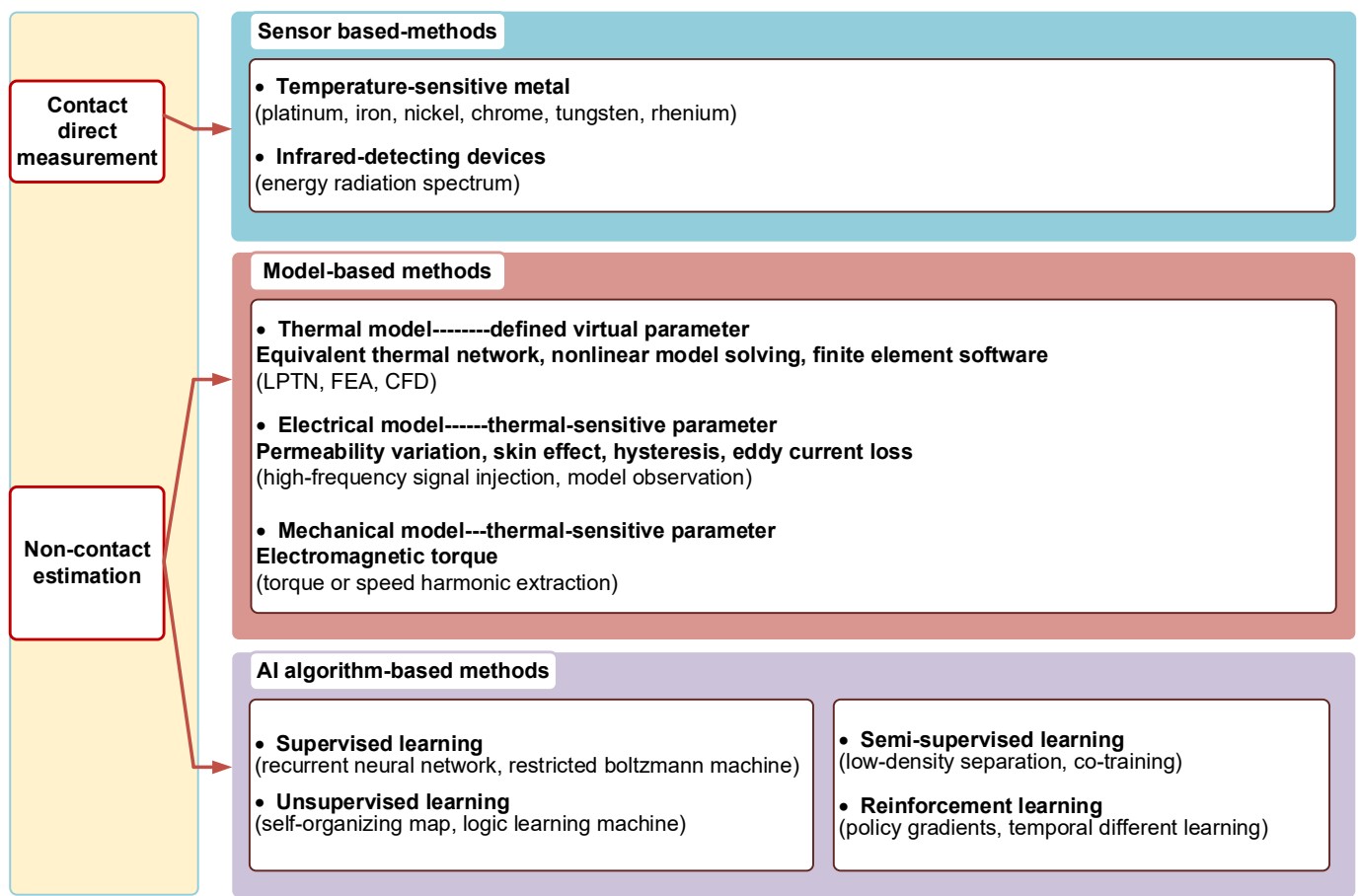

**Figure 1.** Categorization of temperature monitoring methods by techniques.

Sensor-based methods are comprehensible, which are based on temperature sensors. These methods provide the most straightforward way, which is accompanied by the expensive cost and installation problem.

As for the model-based methods, the classification can be based on a thermal model, electrical model, and mechanical model, and the availability of these methods are demonstrated respectively. In contrast to the direct measuring of sensor-based methods, these model-based methods commonly rely on thermal-relevant parameters or are based on the specification and parameters.

Thermal model-based methods directly estimate the temperature. The heat transferring process of machines can be described through the lumped-parameter thermal network (LPTN) models. With temperature information of available positions, the thermal losses are calculated to derive the temperature difference. In addition, finite element methods (FEM) can be used to simulate thermal phenomena.

Electrical model-based methods demonstrate the link between electrical parameters and thermal effect. Due to the different mechanisms between temperature monitoring for the PM and stator windings, electrical model-based estimation methods are introduced separately. For the PM, there are two main methods: signal injection-based methods and PM flux-based methods. Signal injection-based methods are usually implemented by high-frequency signal injection. Through induced high-frequency resistance and inductance,

which are thermal sensitive and vary linearly with temperature, the PM temperature can be derived. The PM flux-based methods also rely on the temperature-sensitive parameter, i.e., the PM flux. Unlike the signal injection, the PM flux can be estimated based on the observational model or direct measurement during normal operation. For the temperature estimation of stator windings, similar to the PM, there are two main methods: observation model-based methods and signal injection-based methods. In comparison with thermal-sensitive parameters of the PM, the resistance of the stator winding is commonly utilized for temperature estimation and the DC signal is usually used for injection.

Mechanical model-based methods exploit the combination of mechanical and electrical parameters in mechanical equations. Through the conjoint analysis of multiparameter relations, the thermal-relevant parameter can be extracted.

In addition, the AI algorithm-based methods used for temperature monitoring are attracting more and more attention and research. From the perspective of model training, these algorithms can be classified into the following categories: supervised learning, unsupervised learning, semi-supervised learning, and reinforcement learning. Relying on advanced processors and excellent adaptability, the AI algorithm-based methods are preferred in extensive monitoring fields.

According to the categorization, various schemes relying on different mechanisms are proposed. Therefore, it is of great significance to review these temperature monitoring techniques for the following purposes.

1.　Provide an introduction of the existing temperature monitoring techniques.
2.　Summarize the monitoring principles of these techniques.
3.　Identify the advantages and limitations.
4.　Discuss the challenges and opportunities for practical applications.

Recently, relevant overview papers have been published to review the temperature monitoring techniques for motor components, such as the PM. In [11], magnet temperature determination techniques are introduced and compared. However, the focus of this paper is on the implementation process. As previously mentioned, a prospective analysis for challenges and opportunities of temperature monitoring and investigation of the research trend is required. In addition, the practical applications should be discussed.

Therefore, the main purpose of this paper is to provide a better understanding and exhibition of the existing methods on feasibility and application prospect. Focusing on the PM and stator windings, the main contribution of this paper lies in the following:

1.　The existing methods for temperature monitoring are categorized and introduced.
2.　The mechanisms of relevant failures caused by temperature rising is presented.
3.　The principles and implementation procedure of the existing methods are introduced.
4.　The advantages and limitations of these methods are summarized and analyzed.
5.　The challenges and opportunities for practical applications are discussed.
6.　The research trend is sketched and presented.

This paper is organized as follows. Section 2 presents the mechanism of temperature rising and thermal-caused failures for the PM and stator windings. Then, as described in the introduction, the detailed introduction of sensor-based methods, model-based methods, and AI algorithm-based methods are demonstrated in Sections 3–5, respectively. The limitations and advantages are presented. In Section 6, possible challenges and opportunities are discussed. In Section 7, the research trend of temperature monitoring is concluded.

## 2. The Mechanism of Thermal-Caused Failures for the PM and Stator Windings

As described in previous analysis, the PM and stator windings are two critical components in PM machines. Therefore, thermal monitoring of the PM and stator windings is essential for reliable operation. Furthermore, appropriate monitoring modes should be determined by the heating mechanism and practical engineering demands.

### 2.1. The heating Sources of PM and Stator Windings

Specifically, non-current excitation is the significant feature of rotor flux in PM machines. However, induced space harmonic magnetomotive force (MMF) can lead to significant eddy currents and incur loss. Therefore, the main cause of PM temperature rise is the eddy-current loss, which can be calculated by [12].

$$P_{eddy} = 2p \sum_{v=1}^{N} \frac{L_{stk}}{2\sigma_m} \int_{-\alpha_p \pi/2p}^{\alpha_p \pi/2p} Re\left[\widetilde{J}_{zv} \widetilde{H}_{\theta v}^*\right] R_m d\theta \tag{1}$$

where $p$ is the pole pairs, $N$ is the maximum order assumed for the eddy current component induced, $L_{stk}$ is the core stack length, $\sigma_m$ is the PM conductivity, $\alpha_p$ is the pole arc to pole pitch, $\alpha_p$ stands for the angular span of the PM over a pole, $R_m$ is the magnet outer radius, in terms of complex quantities, $\widetilde{J}_{zv}$ is the induced current density, and $\widetilde{H}_{\theta v}^*$ is the complex conjugate of tangential magnetic field intensity.

Because of this effect, the eddy-current loss leads to the gradual increase of the PM temperature. In addition, it is worth noting that the frequency of alternating fields has significant effect on the losses.

Furthermore, the mechanical loss can also lead to the temperature rising. Among these factors, air friction loss is the most important and is shown as follows:

$$P_{air} = 3.87 k_{air} \pi \eta_{air}^{\frac{1}{2}} \rho_{air}^{\frac{1}{2}} \omega^{\frac{5}{2}} R_{PM}^3 L \tag{2}$$

where $k_{air}$ is the surface roughness coefficient of the rotor, $\eta_{air}$ is the dynamic viscosity coefficient of the air at one atmospheric pressure, $\rho_{air}$ is the air density, $\rho_{PM}$ is the outer radius of the PM, and $L$ is the rotor length.

Comparatively, stator windings are more prone to overheating. Due to the winding resistance, copper loss is the main reason of winding temperature rising. The copper losses can be calculated as:

$$P_{Cu} = 3I^2 R_{Cu} \tag{3}$$

where $R_{Cu}$ is the phase resistance, and $I$ is the effect value of the phase current.

In addition, there are several other effects due to thermal issues. On one hand, the iron losses from windings and magnetic steel can lead to the temperature rising. The iron loss can be expressed as [13].

$$P_{Fe} = k_{hyst} B_m^{\alpha} f + k_{eddy} B_m^2 f^2 + k_{excess} B_m^{1.5} f^{1.5} \tag{4}$$

where $k_{hyst}$ and $\alpha$ are the hysteresis loss coefficients, $f$ is the frequency, $B_m$ is the amplitude of flux density, $k_{eddy}$ is the eddy current loss coefficient, and $k_{excess}$ is the excess loss coefficient.

On the other hand, rapid increase of temperature may occur due to abnormal condition or system failures, such as an unbalanced load, winding faults, and heat dissipation problems. As a consequence, it is expected that gradual heat accumulation caused by copper loss and generation of short-term hyperthermia may occur during the operation.

### 2.2. Thermal-Caused Demagnetization of PM

The self-magnetization and magnetic domain are basic properties of PM materials. Nevertheless, the magnetism of PM is influenced by the external magnetic field and self-temperature. The practical influence of the external magnetic field is determined by the operating point, which is dependent on the specifications and hysteresis loop. Meanwhile, the trends of the hysteresis loop are influenced by self-temperature. Therefore, thermal stability is essential for operating reliability. The moving of operating point results in the decrease in PM flux density, which leads to the lower capability of output torque. In fact, the operating temperature range is mainly limited in an acceptable torque range. In addition, there is a critical safety margin, of which the upper limit is defined as the Curie

temperature. Although the magnetism of PM is weakened with the temperature rising, the magnetic domain is slightly changed and the magnetism is usually reversible when the maximum temperature does not exceed the Curie temperature. However, once the Curie temperature is reached, the self-magnetization disappears and the structure of magnetic domains is destroyed, which indicates the irreversible demagnetization of PM materials.

The temperature characteristic is dependent on the permanent magnet material. The parameters of some typical permanent magnet materials are given in Table 1. For these materials, $\alpha_B$ is the thermal coefficient, which is defined as the changing rate of PM remanent magnetic flux density with temperature. $B_r$ is the remanent magnetic flux density at normal temperature, which represents the load capacity. The most popular PM material is the NdFeB series and the thermal stability of this material is significantly worse than others.

**Table 1.** The parameters of typical permanent magnet materials.

| PM Material | $\alpha_B$(%/°C) | $B_r(T)$ in Normal | Curie Temperature |
|---|---|---|---|
| Alnico_5 | −0.02 | 1.25 | 530 °C |
| Alnico_9 | −0.02 | 1.05 | 530 °C |
| Sr-Ferrite | −0.2 | 0.3 | 450 °C |
| Ferrite_9 | −0.18 | 0.45 | 450 °C |
| $Sm_2Co_{17}\_35E$ | −0.035 | 1.19 | 820 °C |
| $SmCo_5\_18$ | −0.045 | 0.87 | 820 °C |
| NdFeB_33EH | −0.11 | 1.15 | 310 °C |
| NdFeB_N55 | −0.12 | 1.49 | 310 °C |
| NdFeB_45UH | −0.12 | 1.35 | 310 °C |

The thermal monitoring of PM performs preventative maintenance for an acceptable temperature range. Once the PM temperature reaches the relative threshold value, protective actions should be conducted for stability and safety.

### 2.3. Stator Insulation Aging Caused by High Temperature

The insulation of stator windings in PMSMs contains both inter-turn and groundwall insulation. Insulation aging is accelerated mainly due to thermal stress. To be specific, the chains between the molecules break into small ones during the insulation aging process, and overheating can cause delamination in the insulation. As shown in Figure 2, the capacitance $C_{eq}$ represents the capacitive coupling of the insulation and the resistance $R_{eq}$ represents the dielectric losses in the insulation. The insulation aging caused by overheating is equivalent to the changing of capacitance.

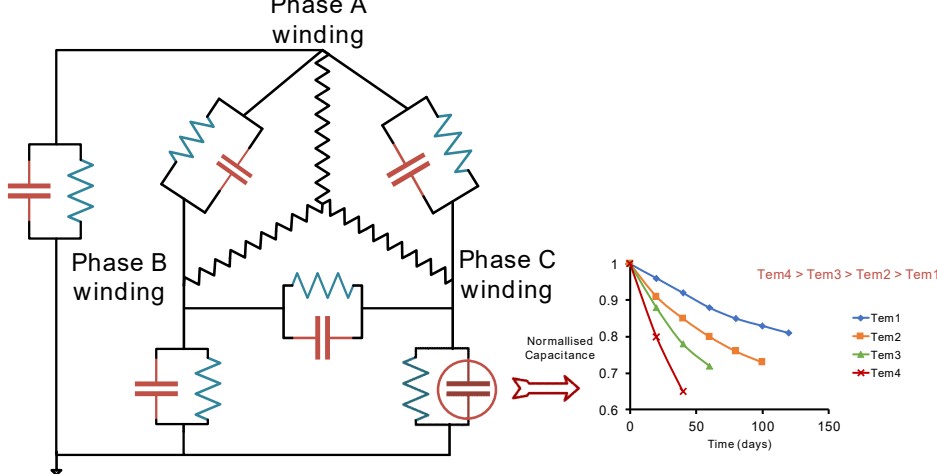

**Figure 2.** Equivalent circuit of the stator winding insulation. Data from [14].

Therefore, the main goal for temperature monitoring of stator windings is to ensure the insulation reliability, which is essential for avoiding damage caused by inter-turn fault.

*2.4. Sectional Discussion*

To sum up, the purpose of temperature monitoring the PM and stator windings is to detect the health condition of components and maintain normal operation. To be specific, except for the overheating caused by the normal operation, faults such as interturn short circuit insulation aging can also lead to temperature rising of stator windings. Therefore, the thermal monitoring of stator windings should be implemented more frequently in response to possible emergencies. Online monitoring is an effective method for confirming the healthy condition of relevant components. Continuous monitoring supports failure prediction. In addition, the insulation aging of hotspots develops faster due to the higher temperature. Therefore, multipoint monitoring is more appropriate than global estimation. As for the PM temperature, considering the relatively uniform temperature rise, the PM temperature should also be monitored in real-time, considering the operating conditions for choosing a reasonable monitoring period. Meanwhile, partial demagnetization caused by nonuniform heating is acceptable due to the inconspicuous temperature difference. Furthermore, installation challenges the multipoint monitoring of temperature monitoring and standardization.

Due to the abovementioned requirements of temperature monitoring, various techniques have been proposed in recent years. These schemes are classified by principles and are presented in the next section.

**3. Sensor-Based Methods of Temperature Monitoring for PM Machines**

Sensor-based monitoring methods are based on direct temperature measurement through thermal-sensitive devices. The basic concept in this category is to obtain the temperature of a target component by precise measurement. These detecting devices are mainly made of thermal-sensitive metals or based on infrared sensing. If permitted, PM machines can be modified to install the temperature sensors. It is convenient to implement direct measurement on stator windings due to the static state. As for the PM, according to the installation manner, the most typical techniques can be classified into battery-powered devices, slip rings, and infrared sensors. Considering that the infrared sensors are additional devices and the machine surface to be tested should be exposed, it is worth mentioning that the infrared temperature measuring methods are categorized as a direct contact measuring method.

Considering that the major challenge of sensor-based methods is located on the PM monitoring, this section is mainly focused on introducing strategies for PM temperature. Generally, the entire temperature monitoring devices are attached on the target component in battery-powered methods. In [2,4], the PM rotor is modified to embed the thermocouples of type K. There are six sensors, of which four are radial surfaced mounted on two ends of the rotor and two are buried in holes drilled in the rotor with a depth of 5 mm and 15 mm, respectively. The two drilled holes are located on the same radial surface on one end. In addition to the modification of the motor, the existing space of the motor is also utilized. Then, the battery devices are integrated to install on the rotor. However, transmitting the temperature information is complex and inconvenient for battery-powered methods. For convenient signal transmission, wireless communication is utilized. In [15], temperature measuring equipment is attached to the PM by inserting it into the extra space. For each PM section, there is a sensor-array that consists of 15 TMP100 sensors in a 3 × 5 arrangement attached to the PM surface. The surface of each PM part is rectangular. These 15 sensors are uniformly distributed in a rectangular array. Meanwhile, the same temperature measurement arrays are configured on the six PM parts, respectively. Nevertheless, attaching sensors to the target component is not always possible in practice. Therefore, no-contact measuring methods based on infrared devices are proposed. In [7], a surface temperature measurement of excitation winding in rotation is presented. The

infrared thermometer used for the measurement is an industrial sensor with a thermopile detector. A measurement junction is connected to a photosensitive element exposed to infrared radiation. Meanwhile, the digital temperature sensors are used for reference. In addition, the infrared thermometer distinguishes two different surfaces in terms of radiation: an excitation winding placed on a salient pole and an interpolar surface. The emissivity factor is set to realize accurate temperature detecting by an industrial infrared camera. Similarly, an IR camera is used to detect the temperature of the rotating surface. In addition, the slip rings can be utilized to transfer the measurement information of PM temperature [16]. The slip rings method is a classical technique utilized in electrical machines. These devices contain stationary and rotating components for transmitting the electrical signal of the rotating sensors.

*Sectional Discussion*

It is observed that these methods all depend on additional devices, which determines the accuracy of the monitoring. Meanwhile, for variable types of machines, the characteristics and actual application requirements are different. Contact measuring is accurate for detecting hot spots, but it is not convenient due to the need for space and modification. In addition, these strategies are complicated when transforming the measuring data. Therefore, these techniques are more applicable in high-capacity and low-speed machines. Non-contact measuring methods are more convenient for application, which require less modification. However, this technique is restricted in practical industrial applications due to its low accuracy. Considering the practical applications in industrial engineering, the sensor-based methods are widely used to estimate the winding temperature due to the relatively static state. Comparatively, the application for temperature monitoring the PM is significantly less restricted by the installation. Thermocouples are fixated to the rotor with cement glue in [2] and a hollow shaft is required to place the PCBs in [15]. These required structural modifications of machines are common in sensor-based methods. The primary concerns of these scholars are the monitoring accuracy and accessibility of temperature distribution. However, the consequent modifications or specific requirements are inevitable. In recent years, PM machines are widely used in automotive drives, which are sometimes miniature and delicate. From the perspective of technique, the availability of sensor-based monitoring methods is limited by the increasing degree of integration. Furthermore, the additional cost is an important indicator to evaluate the feasibility.

## 4. Model-Based Methods of Temperature Monitoring for PM Machines

Restricted by the previously mentioned limitations of sensor-based methods, specifications, available parameters, and measurable signals are utilized in some research to derive the temperature of PM machines. These monitoring techniques rely on typical machine models, which can be classified into model-based methods. As shown in Figure 3, the model-based methods can be categorized into thermal model-based methods, electrical model-based methods, and mechanical model-based methods. For thermal model-based methods, equivalent thermal parameters, such as resistances and capacitances, are calculated to establish thermal network and used to derive the node temperature. In addition, the finite element models can also be used for analyzing the thermal process. For the other model-based methods, relevant temperature-sensitive parameters are respectively utilized for temperature derivation of target components. Therefore, temperature information of target components is directly acquired through thermal model-based methods, and indirectly acquired through the electrical model and mechanical model.

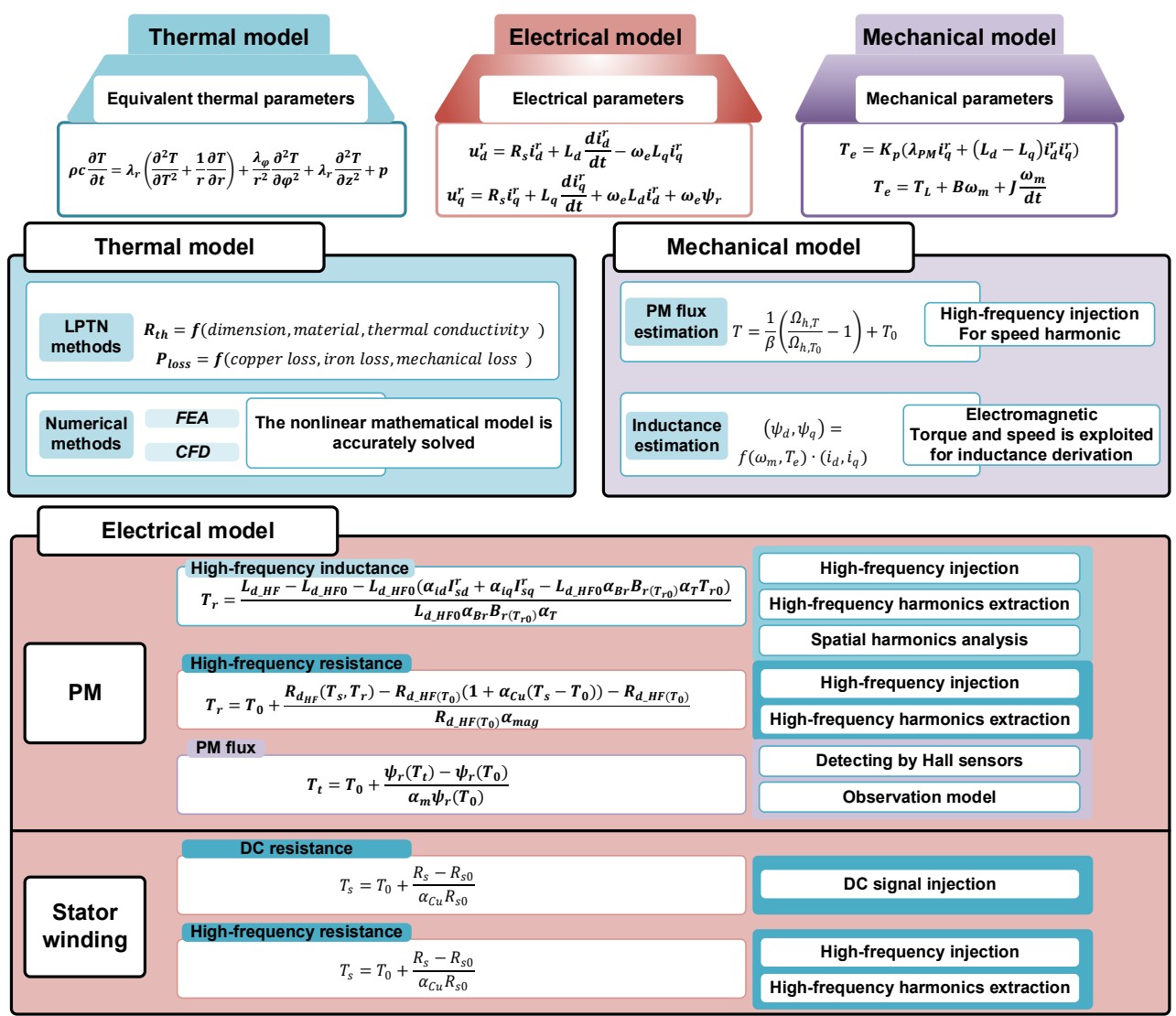

**Figure 3.** Demonstration of specific techniques for model-based methods.

### 4.1. Thermal Model-Based Methods for Temperature Monitoring

Relying on reasonable measurements and specifications, the heat generation and conduction are derived or numerically solved to calculate the temperature in thermal model-based methods. Temperature monitoring techniques for components in PM machines based on thermal models can be classified into parameterized thermal networks and concrete reconstructed simulation. The corresponding typical schemes are LPTNs and FEMs, respectively.

The heat transfer function in cylindrical coordinates ($r$, $z$, $\varphi$) is shown as follows [17]:

$$\rho c \frac{\partial T}{\partial t} = \lambda_r \left( \frac{\partial^2 T}{\partial T^2} + \frac{1}{r} \frac{\partial T}{\partial r} \right) + \frac{\lambda_\varphi}{r^2} \frac{\partial^2 T}{\partial \varphi^2} + \lambda_r \frac{\partial^2 T}{\partial z^2} + p \tag{5}$$

where $\lambda_r$, $\lambda_\varphi$, and $\lambda_r$ are the thermal conductivity coefficient in three directions in the cylindrical coordinate, $\rho$ is the material density, $c$ is the specific heat capacity, $p$ is the internal heat production rate, and $T$ is the temperature of the specific component in a PM machine.

To simplify the analysis, the basic principle of this kind of method is to transfer the partial differential heat equation into a simplified form. The LPTN models are established and rely on the exact thermal energy flow and thermal resistance evaluation. LPTNs

assume that the heat sources are concentrated at the nodes of the thermal network and heat flow is concentrated through network paths. Furthermore, the thermal resistances and thermal capacitances are introduced into the model to establish the heat balance equations. To solve the temperature of the target component, the solution methods for the electrical circuit are exploited. A typical simplified thermal model for the temperature monitoring of rotor magnets is presented in [18]. It presents a network consisting of 12 thermal resistors and 9 thermal nodes. The thermal resistances are calculated based on the dimensions, thermal conductivity, and positions as follows:

$$R_{th} = \frac{\ln\left(\frac{rout}{rin}\right)}{2\pi\lambda L} \times \frac{2\pi}{2p\theta} \tag{6}$$

where $rout$ is the outer radius, $rin$ is the inner radius, $\lambda$ is the thermal conductivity, $p$ is the pole pairs, $L$ is the active length of the machine, and $\theta$ is the angular span.

In addition, the core losses (including both hysteresis and eddy-current iron loss components) are calculated as a function of magnetic flux density and excitation frequency, shown as follows:

$$P = P_0 \left(\frac{B}{B_0}\right)^{\in_0} \left(\frac{f}{f_0}\right)^{\in_1} \tag{7}$$

where $P_0$ is the base power, $B$ is the flux density, $f$ is the frequency, $B_0$ is the base flux density, $f_0$ is the base frequency, and $\in_0$ and $\in_1$ are the flux exponent. $B_0$, $f_0$, $\in_0$, and $\in_1$ are the characteristic parameters of the PM material. The temperature rise between thermal nodes are calculated by the power losses. Then, on the basis of available node temperature, the temperature distribution can be derived. In practice, obtaining comprehensive and accurate parameters of the motor is not always possible. Therefore, simplifying the LPTN for less thermal resistors and nodes is reasonable for accuracy improvement. In this condition, some simplified thermal network models are proposed. In [19], relying on the available stator core temperature, a thermal model with three thermal nodes is presented. The three thermal nodes represent the PMs, the stator winding, and the stator core, respectively. The thermal conductance and capacitances are obtained from a transient reference set of temperatures. $T_{pm}$ and $T_s$ represent the temperature of the PM and stator winding, respectively. Specific to the preliminary implementation, the temperature of the components should be sensed in advance. The stator core temperature $T_c$ is measured and used as the input quantity in this thermal network and the ambient temperature $T_{ref}$ is not required. The total stator winding losses and heat flow source $P_p$ can be calculated by means of

$$P_s = 3R_s I^2, \frac{P_p}{P_{p,ref}} = \left(\frac{f}{f_{rated}}\right)^a \times \left(\frac{I}{I_{rated}}\right)^b \tag{8}$$

where $R_s$ is the winding resistance, $P_{p,ref}$ is the reference losses of PM, and rotor core obtained from a 2D FEA, $f_{rated}$ and $I_{rated}$, are rated stator frequency and phase current, respectively, and the parameters $a$ and $b$ are results of best fit analysis. It worth mentioning that the winding resistance is temperature dependent, which is determined by the following equation:

$$R_s = 3R_{s,20}(1 + \alpha_{Cu}(R_{s,20}(T_s - 20\,°C))) \tag{9}$$

where $R_{s,20}$ is the winding resistance at 20 °C, and $\alpha_{Cu}$ is the temperature coefficient of copper. Based on the thermal structure and power losses, the temperature of the PM and stator winding can be derived through one measured temperature.

To sum up, there are two main issues for thermal model-based methods: the structure design and parameter calibration. On the one hand, the manner of heat transfer is too complex to accurately describe using simple electrical circuits. However, the precision of the available component temperature is restricted by the sensors. Furthermore, thermal phenomena, in specific conditions, is not exactly the same as the empirical derivation. In this

condition, for more straightforward thermal simulation, directly solving thermodynamic equations can be utilized.

The FEM analysis is implemented based on computer-aided engineering tools, such as the ANSYS. The heat transfer and fluid functions are precisely established with boundaries of temperature and thermal flux. Meanwhile, the numerical calculation based on flux simulation, such as computational fluid dynamic (CFD), can be conducted to obtain accurate temperature information. In [20], a 3D-FEM model of PM is employed for thermal analysis. The temperature distribution of the machine is investigated and obtained. Two PMSMs with different shapes of PM are studied through FEM models in [21]. Combining 2D and 3D Finite Element models, the thermal effect of the water jacket and the maximum temperature of the stator windings are depicted. Considering that the complicated operating conditions cannot be accurately simulated through the FEM method, temperature monitoring is seriously affected by the simulation accuracy of the machine body and operating conditions.

Sectional Discussion

In fact, through precise numerical analysis, the thermal distribution of components in a machine can be acquired theoretically. As discussed in [17], the LPTN methods are not outstanding in applicable operating areas. In [18], possibilities of LPTNs are listed as transient thermal analysis and convection cooling. Transient thermal analysis can be utilized as a prediction technique, which is significant for protection. Nevertheless, compared with the LPTN methods, results obtained from the FEM are more comprehensive and detailed. Transient prediction through FEM methods is more reliable. However, the computational complexity is obviously higher than the LPTN methods, which affects the real-time capability. Meanwhile, the applicability is restricted by the requirement of a specific dimension. As for the LPTNs, node temperature is sufficient for global thermal analysis of machine components. Nevertheless, the structure and parameter calibration are critical issues for accurate simulation, which inevitably requires pre-tests and additional thermal information.

*4.2. Electrical Model-Based Methods for Temperature Monitoring*

The electrical model of PMSM is the theoretical foundation of large number of research, such as parameter identification and optimization control. Generally speaking, on account of the decoupling and parameter stability, the electrical model of a PM machine in the synchronous rotor reference frame is utilized for analysis. These equations are expressed as follows:

$$
\begin{aligned}
u_d^r &= R_s i_d^r + L_d \frac{di_d^r}{dt} - \omega_e L_q i_q^r \\
u_q^r &= R_s i_q^r + L_q \frac{di_q^r}{dt} + \omega_e L_d i_d^r + \omega_e \psi_r
\end{aligned}
\tag{10}
$$

where $u_d^r$ and $u_q^r$ are $d$- and $q$-axis stator voltages in the rotor reference frame, and $i_d^r$ and $i_q^r$ are $d$- and $q$-axis stator currents in the rotor reference frame. $R_s$ is the stator resistance, $L_d$ and $L_q$ are $d$- and $q$-axis inductances, $\omega_e$ is the electrical angular velocity, and $\psi_r$ is the PM flux linkage.

As demonstrated above, the main electrical parameters are involved in this mathematical model. The basic principle of temperature monitoring methods based on the electrical model is to monitor the temperature-sensitive parameters. Due to the different temperature-sensitive parameters, there are significant distinctions between the temperature monitoring techniques for the PM and stator windings. Therefore, techniques for different components are introduced respectively.

4.2.1. Monitoring Techniques for PM Temperature

In order to minimize the impact on operating stability, realizing the parameter estimation during normal operation is preferred. However, the electrical mathematical model is a strongly nonlinear relationship that contains four changeable parameters. In this condition,

the nonlinear relationship and rank-deficient problem limit the accuracy of parameter estimation using fundamental-frequency models.

Therefore, in recent research papers, high-frequency parameters used to estimate the PM temperature are resistance and inductance. In fact, these temperature monitoring techniques are essentially dependent on parameter identification.

In typical inverter drive systems, the switching devices are turned ON and switched OFF at a high frequency, which generates high-frequency harmonics. Meanwhile, the sinusoidal signals corresponding to the rotation mainly contain the fundamental-frequency. Therefore, the high-frequency and fundamental-frequency signals are involved in measurable quantities, such as voltages and currents. However, the amplitude of high-frequency signals is not sufficient to extract and exploit. Therefore, for enhancing the dominance of high-frequency signals, techniques based on a signal injection are proposed in some papers. Meanwhile, thermal-sensitive parameters, such as the PM flux, can also be detected by fundamental-frequency signals. Therefore, the main techniques for PM temperature monitoring can be classified into invasive signal injection-based methods and noninvasive PM flux-based methods.

A.    Signal injection-based methods for PM temperature estimation

In order to distinguish these techniques, relevant thermal-sensitive parameters can be categorized into high-frequency parameters and fundamental-frequency parameters.

The purpose of signal injection for thermal monitoring of the PM is to establish the high-frequency models for tracking and extracting the temperature-sensitive parameters, such as the high-frequency resistances and inductances. The correlation between these high-frequency parameters and the PM temperature has been demonstrated in the literature.

Theoretically, the stator inductances in the $d$- and $q$-axis are determined by the saturation level. Considering that the fundamental-frequency currents are dominant, the saturation level is determined by the fundamental currents. Combined with the effect of a magnetic field from the PM, the inductances are dependent on the fundamental-frequency currents and the PM remanent flux. Therefore, with the known currents, the thermal condition of PM can be reflected in inductance.

Considering that no high-frequency component containing the PM flux-term and injecting frequency is sufficiently higher than the rotation frequency, the PM flux-term $\omega_e \psi_r$ can be neglected. Therefore, a high-frequency model can be obtained from a fundamental-frequency model as follows.

$$
\begin{aligned}
u_{d\_HF}^r &= R_{d\_HF} i_{d\_HF}^r + L_{d\_HF} \frac{di_{d\_HF}^r}{dt} - \omega_e L_{q\_HF} i_{q\_HF}^r \\
u_{q\_HF}^r &= R_{q\_HF} i_{q\_HF}^r + L_{q\_HF} \frac{di_{q\_HF}^r}{dt} + \omega_e L_{d\_HF} i_{d\_HF}^r
\end{aligned}
\tag{11}
$$

where $u_{d\_HF}^r$, $u_{q\_HF}^r$, $i_{d\_HF}^r$, and $i_{q\_HF}^r$ are stator $d$- and $q$-axis high-frequency voltages and currents in the rotation synchronous reference coordinate, respectively. $R_{d\_HF}$, $R_{q\_HF}$, $L_{d\_HF}$, and $L_{q\_HF}$ are the $d$- and $q$-axis high-frequency resistances and inductances, respectively.

In [22], the high-frequency inductance of the $d$-axis is used for PM temperature estimation. The expression of high-frequency inductance can be derived as follows:

$$
L_{d\_HF} = L_{d\_HF0} \left( \alpha_{id} I_{sd}^r + \alpha_{iq} I_{sq}^r + \alpha_{Br} \left( B_{r(T_r)} - B_{r(T_{r0})} \right) \right)
\tag{12}
$$

where $I_{sd}^r$ and $I_{sq}^r$ are fundamental-frequency currents of the $d$- and $q$-axis, $B_{r(T_{r0})}$ is the PM remanent flux at room temperature $T_{r0}$, $B_{r(T_r)}$ is the PM remanent flux at arbitrary temperature $T_r$, $\alpha_{id}$, $\alpha_{iq}$, and $\alpha_{Br}$ are coefficients linking high-frequency inductance with fundamental current in the $d$-axis, high-frequency inductance with fundamental current in

the *q*-axis due to cross-coupling, and the PM remanent flux, respectively. Considering the thermal characteristic of the PM, the temperature can be derived as follows:

$$T_r = \frac{L_{d\_HF} - L_{d\_HF0} - L_{d\_HF0}\left(\alpha_{id}I_{sd}^r + \alpha_{iq}I_{sq}^r - L_{d\_HF0}\alpha_{Br}B_{r(T_{r0})}\alpha_T T_{r0}\right)}{L_{d\_HF0}\alpha_{Br}B_{r(T_{r0})}\alpha_{PM}} \tag{13}$$

where $\alpha_{PM}$ is the thermal coefficient of the PM.

Meanwhile, the other relevant coefficients in this equation should be obtained experimentally. Then, the PM temperature can be acquired.

Similarly, high-frequency resistance can also be used to obtain the PM temperature. In general, winding resistance is considered as an intrinsic quality of conductor materials, which stays constant at a fixed temperature. It is confusing to put forward the concept of high-frequency resistance. However, the skin effect and proximity effect occur when high-frequency currents flow through the conductor. The changing current density causes an increase in heating power. Moreover, for the PM, due to the high-frequency injection, hysteresis and eddy current losses are produced. In this condition, the aforementioned phenomenon leads to additional energy converting into thermal losses. Therefore, the high-frequency effect is reflected in stator winding and PM, which can be represented by defining high-frequency resistance. Considering the temperature characteristic, the overall high-frequency resistances in the *d*- and *q*-axis can be subdivided into contributions of stator and rotor, which can be expressed as follows:

$$R_{d_{HF}}(T_s, T_r) = R_{d_{HF\_s}}(T_s) + R_{d_{HF\_r}}(T_r) \tag{14}$$

where $R_{d_{HF\_s}}(T_s)$ is the winding component at temperature of $T_s$ and $R_{d_{HF\_r}}(T_r)$ is the PM component at temperature $T_r$. In this condition, the PM component of high-frequency resistance can be extracted to estimate the PM temperature. In [23], a PM temperature estimation technique is proposed based on high-frequency resistance. The estimation principle can be derived as follows:

$$T_r = \frac{R_{d_{HF}}(T_s, T_r) - R_{d\_HF(T_0)}(1 + \alpha_s(T_s - T_0)) - R_{d\_HF(T_0)}}{R_{d\_HF(T_0)}\alpha_{PM}} \tag{15}$$

where $\alpha_s$ and $\alpha_{PM}$ are the thermal coefficients of stator winding and the PM, and $T_0$ and $T_s$ are the initial temperature and current temperature of the stator winding.

With the available stator winding temperature detected by thermal sensors, the PM temperature can be derived.

In addition, in [24], a voltage pulse is applied in the *d*-axis while the *d*-axis current is measured. Through voltage pulse injection applied in the winding of phase A when the *d*-axis is coincident with the winding axis of phase A, the voltage of the *d*-axis is converted to $\frac{2}{3}V_{dc}$. Then, the PM temperature can be derived based on the current slope, which represents the magnetization level of the PM.

In [25], analysis shows that the dynamic permeability depends on the core saturation level. Meanwhile, the core at the a-axis is less saturated as the PM temperature increases, which influences dynamic permeability. In addition, the DC component of high-frequency inductance can be derived using a winding function and dynamic permeance. Therefore, by considering the effect of spatial harmonics on the high-frequency inductance, the PM temperature can be calculated and calibrated.

To sum up, the signal injection techniques for PM temperature monitoring are implemented based on the electrical mathematical model of high-frequency parameters. Utilizing the effects caused by high-frequency signals, corresponding impedance is derived by available electrical signals to extract the PM temperature. In practice, high-frequency pulsating or sinusoidal signals can be used for this purpose. Highly precise monitoring can be achieved by signal injection methods. Meanwhile, no additional devices are required in most cases. However, extra torque ripple of specific-frequency will be introduced during

the injecting process. To guarantee that the specific-frequency currents circulate, the control module or algorithm should be modified, which may not be available in some highly integrated devices.

B.    PM flux-based methods

As introduced in Section 2, the performance of the PM installed on the rotor using surface-mounted or embedded modes is affected by the temperature. The remanent flux density is linearly related to temperature. Considering that the hysteresis loop is straight in the operating area, the PM flux density of the working point varies linearly with temperature. Then, the PM flux can be regarded as a thermal-sensitive parameter. In most cases, it is sufficient to express the estimation principle as follows.

$$\alpha_{PM} = \frac{B(T_t) - B(T_0)}{B(T_0)(T_t - T_0) \times 100\%} \Rightarrow T_t = T_0 + \frac{\psi_r(T_t) - \psi_r(T_0)}{\alpha_{PM}\psi_r(T_0)} \tag{16}$$

where $\alpha_{PM}$ is the thermal coefficient, $T_0$ and $T_t$ are the temperatures of the initial moment and the subsequent moment, $B(T_0)$ and $B(T_t)$ are the PM flux density at corresponding moments, and $\psi_r(T_0)$ and $\psi_r(T_t)$ are the PM flux at corresponding moments.

As shown in Equation (6), PM flux is involved in the back electromotive force (back-EMF) term of the electrical mathematical model, which is available for noninvasive PM temperature monitoring. Therefore, PM flux monitoring for rotor temperature estimation is studied in some papers. In [26], the hall sensors originally installed in the PMSMs for initial position detecting are used for the PM flux measurement. The analog signal provided by the hall sensors is sensitive to the PM flux density, which can be utilized for the temperature estimation. It is worth mentioning that this technique is not exactly based on the electrical model. However, considering that the thermal-sensitive parameter is used, this method is subsumed under this category.

Except for the direct measurement method, the PM flux can be estimated through solving the mathematical functions. For this purpose, observation models are commonly used to identify the PM flux, such as the recursive least square (RLS), affine projection algorithm (APA), and extend Kalman filter (EKF). In general, the implementation of these estimation methods is based on the steady state model equations. Considering the thermal or saturation characteristics, several parameters involved in the model are influenced by the operating conditions. To be specific, the PM flux and stator winding resistance are influenced by the component temperature, and stator inductances in the *d*- and *q*-axis are fluctuated with load condition. Therefore, there are four unknown parameters that should be estimated based on the electrical model, theoretically. As a consequence, the estimation accuracy is limited by the rank-deficient problem. In this condition, there are increasing research papers focused on solving the rank-deficient problem. In [27], the EKF algorithm is used to estimate the PM flux, which is implemented as the inductance, assumed to be constant. In [28], a two-time scale dynamic model is proposed to make the machine's full rank and simultaneously estimate the parameters. Considering the difference in the dynamic of the variable parameters, two APAs with different convergence rates are combined. Similarly, two RLS models on different time scales are presented in [29] for parameter identification. In [30], an observation model is proposed on the basis of extensive measurements and data storage in a lookup table (LUT). In these methods, available quantities are estimated and compared with measured values, which is used for calibration of PM. With the update of measured signals, a cyclic iteration algorithm is established to constantly optimize the estimating result. The iterative logic is shown as follows:

$$\hat{\psi}_r(k+1) = \hat{\psi}_r(k) + \mathbf{W}(\mathbf{y} - \hat{\mathbf{y}}) \tag{17}$$

where $\mathbf{W}$ is the gain matrix, $\mathbf{y}$ is the matrix of measured quantities, $\hat{\mathbf{y}}$ is the matrix of estimated quantities, $\hat{\psi}_r(k)$ is the estimated PM flux of a previous iteration, and $\hat{\psi}_r(k+1)$ is the estimated PM flux of this iteration.

In fact, these algorithms are mainly dependent on the previously mentioned recursive and iterative logic. During these strategies, the normal operation is not interrupted and no additional information is required, which shows extensive applicability. However, due to the simplicity of derivation and measuring deviation, the temperature estimation accuracy in these methods is not superior.

4.2.2. Monitoring Techniques for Temperature of Stator Windings

The typical material used for stator windings for PMSMs is copper, which has resistance that is linearly dependent on the temperature. Therefore, the temperature estimation of stator windings based on the electrical model relies on winding resistance. Similarly, variable-frequency resistances are able to estimate the winding temperature. As introduced in the PM temperature monitoring techniques, high-frequency resistance can be generated by signal injection [31]. However, to avoid the interference of hysteresis and eddy current losses caused by the PM, fundamental-frequency or DC resistance is preferred for parameter identification. It is worth noting that the thermal resistive coefficients utilized for the PM and winding resistance are not consistent. For the winding temperature estimation based on resistance at a different frequency, the derivation can be expressed as follows:

$$T_s = T_0 + \frac{R_s - R_{s0}}{\alpha_{Cu} R_{s0}} \tag{18}$$

where $T_0$ and $T_s$ are the initial and current temperatures, $\alpha_{Cu}$ is the thermal resistive coefficient of copper, $R_{s0}$ and $R_s$ are the stator winding resistances at temperatures $T_0$ and $T_s$. It worth mentioning that the thermal relevant equation is suitable for both DC and alternating frequency components.

Similarly, the estimation techniques can be classified into noninvasive observer-based methods and invasive signal injection-based methods.

A.    Noninvasive observer-based methods

Similar with the PM flux, the estimation of winding resistance based on the electrical model is essentially parameter identification. The implementation of observation is limited by the ill-convergence of parameter estimation due to rank-deficient state equations of PMSM. Therefore, to establish a suitable and feasible model for parameter estimation, the satisfaction of required properties should be ensured. Multiple parameters are estimated based on a nonlinear interconnected observer in [32]. In [33], the model reference adaptive system (MRAS) is used to estimate the winding resistance.

B.    Invasive signal injection-based methods

The DC signal is commonly selected to facilitate the distinguishing of injecting signals and fundamental-frequency signals. Meanwhile, compared with a high-frequency injection, the skin effect of the stator windings will not be generated, which reduces the heat emission. Meanwhile, the hysteresis and eddy current losses of the PM are also mitigated.

Intermittent current injection based on the rotating coordinates is presented in [34] for resistance monitoring, and by regulating the $q$-axis injection current, the torque pulsation caused by injection is reduced. Meanwhile, the injection current and induced voltage are used to calculate the winding resistance.

4.2.3. Sectional Discussion

In most literatures discussed in this section, temperature monitoring of a specific part usually requires the temperature of other parts or accurate parameters. Parameters are coupled in the electrical model. For example, in [22], previous knowledge of inductance and decoupling of current effects are required. The authors tend to acquire relevant information through preliminary experimental tests. Furthermore, the reference temperature of the target component cannot be convinced with a consistent standard. In [22], the thermocouples are attached to the rotor. However, the hot spots of the specific test machine are dependent

on the structure and operating conditions. The conflict between global estimation and practical uneven temperature distribution affects the temperature calibration.

The main limitation and drawback of parameter identification-based methods for temperature monitoring is the accuracy. The parameter estimation error is scaled up by temperature coefficient. For example, as shown in Table 1, $\alpha_B$ is $-0.11\%/°C$ for NdFeB 33EH, which means that the PM flux estimation error will be approximately magnified by 900. Therefore, accurate PM temperature monitoring requires extremely high precision for the estimation of PM flux. However, high estimation precision is restricted by the inherent problems of parameter identification. In fact, due to the requirement of extensive electrical parameters, deviation of the modelling establishment is incidental and inevitable. Furthermore, considering the complete dependence of electrical model, the voltage source inverter (VSI) nonlinearity affects the precision of methods based on parameter identification. Therefore, the accuracy of estimation based on the fundamental-frequency signals is susceptible. Signal-injection provides significant improvement in precision, but the hysteresis and the eddy current losses will be increased due to the high-frequency components.

In addition, the temperature monitored by the thermal-sensitive parameters is the global temperature of the PM or stator windings, which is not suitable for the detailed detecting of local hot spots.

### 4.3. Mechanical Model-Based Methods for Temperature Monitoring

The mechanical model combines electrical signals and mechanical signals, which provides the possibility for the transitive estimation of thermal parameters by motion parameters. In general, the mechanical model means the torque equations, which can be expressed by electrical and mechanical parameters, are as follows:

$$
\begin{aligned}
T_e &= K_P\left(\psi_r i_q^r + (L_d - L_q)i_d^r i_q^r\right) \\
T_e &= T_L + B\omega_m + J\frac{d\omega_m}{dt}
\end{aligned}
\tag{19}
$$

where $T_e$ represents the electromagnetic torque, $K_P =1.5P$ and P is the pole pairs, $T_L$ is the load torque, $B$ is the friction coefficient, $J$ is the moment of inertia, and the other parameters are described in previous equations.

With PM flux involved in this model, the mechanical parameters of the machine can be utilized for the PM temperature estimation when combining the two equations. In [35], a linear temperature model is presented to demonstrate that the speed harmonic decreases linearly with the increase of PM temperature, which is derived from the PMSM mechanical model. Suitable harmonic currents satisfying certain requirements are injected into the machine to induce speed harmonics. Based on the linear relation between PM temperature and PM flux, the PM temperature can be estimated by following equation:

$$
T = \frac{1}{\beta}\left(\frac{\Omega_{h,T}}{\Omega_{h,T_0}} - 1\right) + T_0
\tag{20}
$$

where $\beta$ is the PM thermal coefficient, $\Omega_{h,T}$ and $\Omega_{h,T_0}$ are the measured speed harmonic when the PM temperature is $T$ and $T_0$, respectively.

Sectional Discussion

Temperature monitoring based on the mechanical model relies on the rotation speed and several mechanical parameters. In [35], the friction coefficient and moment of inertia are considered as fixed quantities. The typical motor drive system is complicated, and the mechanical parameters may fluctuate with the operating condition. Meanwhile, the extraction of high-frequency mechanical signals is more prone to be disturbed by operating conditions or irresistible vibrations. Theoretically, utilizing the mechanical parameters for PM temperature monitoring is an identification of PM flux. Compared with electrical model-based methods, the rank-deficient problem is circumvented. In addition, specific to high-quality automation drives, thermal protection is critical, and high estimation

accuracy can be obtained through this technique based on the high-precision device for position sensing.

However, there are some drawbacks for this kind of technique. On the one hand, the speed harmonics are more difficult to generate and extract at higher running speed. On the other hand, the main drawback of this method lies in the torque fluctuation caused by the injecting harmonic currents.

*4.4. Discussion for Model-Based Methods*

To sum up, the typical models utilized for temperature monitoring are artificially defined, such as LPTNs or inherent mathematical models, such as electrical functions. The inherent models are more comprehensive and the defined models are more straightforward. In terms of technical complexity and estimation accuracy, methods requiring additional devices or a numerical solver have higher precision. Oppositely, methods aiming to estimate temperature during normal conditions are paying more attention to simplicity and convenience. Overall, compared with the sensor-based methods, these techniques based on models are less invasive.

**5. AI algorithm-Based Methods of Temperature for PM Machines**

Artificial intelligence (AI) is an emerging technique that branched out of computer science. Human intelligence is simulated and extended in this area of research. Meanwhile, this technique is based on the algorithm, data, and computing ability. It is worth mentioning that this section focuses on monitoring techniques that mainly rely on the AI algorithm. Machine learning is the core technique in these methods. Through training in historical data, implicit information can be extracted by the constructed network. The main advantages of the AI algorithm-based methods are the extensive applicability and automaticity. Compared with the model-based methods, the precision limitations caused by the required expertise and sophisticated modelling are avoided. Meanwhile, the estimated temperature is derived by measurable quantities and empirical properties involved in the historical data set. The properties reveal the relation between measurable quantities and target parameters. Furthermore, the AI techniques guarantee the robustness for a variety of operating conditions. Common AI techniques for temperature monitoring are neutral networks (NN), Particle Swarm Optimization (PSO), genetic algorithm (GA), etc.

In general, the AI algorithm-based methods are based on collecting extensive measurements of quantities and target parameters, which are independent of the motor specifications and mathematical models. These quantities should be electrical, mechanical, and thermal parameters. Absolutely, measurable thermal quantities are usually necessary for PM temperature monitoring. In the meanwhile, the condition-relevant quantities are also indispensable. Through adequate training based on data-driven algorithms, the updated measurements can be used to derive the updated parameters, which are unavailable by direct measuring. For temperature monitoring of PM machines, AI algorithm-based methods are getting attention in recent years. To be specific, the temperature of target components and other relevant available signals should be recorded and taken as inputs for training. Taking the deep recurrent and convolutional neural networks presented in [36] as an example, there are sequence learning and multilevel calculating processes involved in this algorithm. Therefore, based on this construction, this algorithm can be called multilayer perceptron (MLP).

The architectures of both topologies are both based on the MLP algorithm. These layers can be classified into the input layer, hidden layer, and output layer. There are weights between layers. The node values in one layer are updated gradually. Finally, the output sequence is obtained and the weights and other parameters can be modified by the estimation errors. In this condition, adequate data is needed for better performance.

The sequence learning mode of a state-of-the-art ANN technique called the temporal convolutional network (TCN) is shown [36]. It inherits recent advances of applications on sequential data and dilated convolutions of the convolutional neutral network (CNN).

The prediction $\hat{y}_T$ is informed by the corresponding observation $x_T$ and a finite set of past observations. The input sequence consists of the external temperature, measurable electrical parameters, and rotor speed. The targets are the temperatures of the PM, stator teeth, stator winding, and stator yoke. The function of hidden layer can be concluded as the data abstraction, which extracts the correlation patterns of inputs and passes the results to the next layer. The hidden layers are generated by repeating updates to acquire acceptable errors between output and sample values. The targets are the thermal quantities of the stator components and the PM.

Meanwhile, the observations can be distributed apart from each other over the sequence rather than being adjacent. In this mode, the crucial, further past events can be detected for reasonable training. With causal and dilating temporal convolutions, this dilated operation leads to an effectively expanded receptive field of the TCN, which would be directly proportional to the network's depth.

In [37], a Nonlinear Auto-Regressive with eXogenous-input (NARX) model based on a difference-estimating feedforward neural network (DFNN) is used to monitor the temperature of the PM and windings of PMSMs. The NARX handles the nonlinear relationships between the state at the next moment and the past information. In contrast to the traditional neural network, this paper proposes to introduce a scalar transformation coefficient to the feedback chain. The training is conducted with an input layer containing the delayed temperature values of five parts and the recorded initial condition. The switching function, measured currents, machine position, voltage output of the current controller, estimated speed, and measured ambient temperature are recorded as the initial input vector. In addition, the temperature of PM and stator windings are measured as the initial state vector.

*Sectional Discussion*

The AI-based methods within the scope of this section all belong to the category of machine learning. AI-based methods have been the research trend in the thermal monitoring field. Similarly, the available thermal quantities and electrical quantities are necessary for training. The training strategies are unique for different AI methods. In [38], several machine learning models are empirically evaluated on their estimation accuracy for the temperature monitoring. The feasibility of these methods is verified in this paper. The scholars thought that temperature monitoring could be implemented through recorded test data without relevant expertise. Indeed, AI-based methods provides a convenient strategy for thermal monitoring. Compared with previous methods, no additional devices and specific parameters are required in AI algorithm-based methods. Focusing on accuracy or simplicity as the priority is an alternative in these algorithms. Meanwhile, the improvement of controllers supports data measuring and algorithm computing. The main problem of these techniques is that real-time capability is limited by data training and multilayer processing. The application for different machine types should be further demonstrated.

## 6. Discussion for Challenges and Opportunities

The temperature monitoring of PM machines has been researched for several decades. Relevant techniques are widely proposed and optimized. However, due to the complex dynamic characteristics of the motor system, there are several problems and conflicts remaining to be solved. Meanwhile, the possible opportunities in the future are discussed.

### 6.1. Challenges for Present works of Temperature Monitoring

6.1.1. The Non-Linearity of Thermal-Sensitive Parameters

In general, the relation between component temperature and relevant thermal-sensitive parameters is supposed to be linear. However, the variation of thermal-parameters are not only influenced by the temperature. For example, when the PM temperature is close to the Curie temperature, the magnetic intensity decreases sharply and therefore the variation of PM flux based on temperature is not really linear. Therefore, more precise relation is

required for temperature monitoring. In this condition, the preliminary temperature test or more temperature factors must be conducted or considered.

### 6.1.2. The Conflict between Monitoring Accuracy and Technical Complexity

As presented in the previous research overview, high estimation accuracy always represents high technical complexity. PM flux observer-based methods can be implemented during normal operation with typical controllers, but the estimation accuracy is susceptible to conditions. With sufficient precision, direct measuring requires additional devices, structure modification, and significant processing time are essential for FEM models and AI algorithm. Comparatively speaking, the AI algorithm-based methods provide alternatives between estimation accuracy and real-time capability. However, in typical industrial applications, the required computing ability is hard to achieve for real online estimation.

### 6.1.3. The Conflict between Anti-Interference Capability and Real-Time Capability

Most online monitoring methods with excellent real-time performance are totally dependent on electrical signals, such as electrical model-based methods. Nevertheless, the temperature derivation based on electrical signals is susceptible to electromagnetic interference (EMI), which mainly contains conductive interference of information and the conductive interference of electronic noise. For thermal-model and AI algorithm methods, the modelling or training based on consistent monitoring and immunity is more robust for temperature estimation. However, the processing time for FEM models and AI algorithms is too long to realize real-time temperature monitoring. By comparison, even though sensor-based methods are limited in many conditions, this approach is more harmonious in both aspects.

### *6.2. Opportunities for Future Works of Temperature Monitoring*

### 6.2.1. Combination of Multi-Type Techniques for Temperature Monitoring

The combination of techniques provides the possibility of complementary advantages. Typical techniques cannot all be feasible in extensive operating conditions. For example, at low speeds, the performance of PM flux observer-based methods is poor and signal-injection-based methods are more robust. Therefore, a combination of these two methods provides less invasiveness and more accuracy. In addition, the installation of thermal sensors can be guided by FEM analysis, which provides extensive temperature distribution.

### 6.2.2. Prospects of System-Level Temperature Monitoring of PM Machines

In general, the temperature distribution is nonuniform due to different thermal properties and heat generation. Therefore, existing temperature monitoring methods are usually focused on individual components. However, thermal protection is required for all critical components in a PM machine, such as the PM and stator windings. Improving the 0 degree of the integration and miniaturization of the drive system, the overall temperature situation of PM machines is necessary and crucial. Therefore, the prospect of system-level temperature monitoring is introduced in the section.

All these techniques have the potential of system-level temperature monitoring. In terms of technique, sensor-based methods are convenient for temperature distribution monitoring and is sufficient at detecting hot spots. However, the increased cost and installation difficulty are the main limiting factors. Model-based methods relying on parameter identification have the potential to conduct multiparameter estimation, such as the node temperature, the PM flux, stator winding resistance, and inductances, all of which are indicators of component temperatures. Nevertheless, only specific components can be monitored by thermal-sensitive parameters and the estimation accuracy is limited. Theoretically, the most feasible techniques for reliable system-level temperature monitoring are LPTNs, numerical methods, such as FEMs, and AI algorithm-based methods. The common characteristic of these techniques is the capability to supply sufficient thermal

spots, and the mapping of the machine temperature can be realized through accurate-specification modelling or preliminary monitoring.

## 7. Research Trend for Extensive Monitoring for PM Machines

In fact, there are more literatures focused on the temperature monitoring of PM machines. The references [39–60] belong to the thermal model-based methods. References in [61–67] exhibit methods based on the high-frequency injection for PM flux monitoring. References [68–75] are methods for PM flux monitoring without signal injection. References [76–88] are based on the resistance estimation for the temperature monitoring of PM machines. [89] relies on the high-frequency inductance for temperature estimation. References [90–92] belong to the AI-based methods. [93–99] are literatures about the characteristics of PM materials. [100–104] studied the thermal losses and aging of electrical machines. [105–107] are about the identification of thermal-relevant parameters. [108,109] are other reviews about the thermal monitoring of electrical machines. [110–114] are some literatures for the researches referenced by this review. In the meanwhile, to demonstrate the development of existing methods, reference papers are summed up in Figure 4. The direct monitoring targets are taken as the distinction. Although the high-frequency resistance and winding resistance are indicators of different components, these techniques are in the category considering the technical overlap. According to the timeline, the research trend in recent years for temperature monitoring is focused on the thermal model methods, PM flux estimation methods, and AI algorithm methods. This research trend is related to the technical extensibility of these methods. Except for the aforementioned system-level temperature monitoring, condition monitoring is also critical. Temperature monitoring and condition monitoring of the PM can both be achieved just by PM flux estimation. In addition, the comparison of these temperature monitoring techniques is demonstrated in Table 2. Requirements and performance for the implementation of these methods are listed. Different focuses for temperature monitoring in industrial applications lead to different suitability of these techniques. In the opinion of the authors, the current research trend originated from the requirement for extensive monitoring of machines. Moreover, the extensive system-level monitoring for either temperature or condition is indeed of great significance for reliable operation and maintenance.

**Table 2.** Technical comparison of temperature monitoring methods.

| Target | Monitoring Techniques | | Computational Complexity | Precision | Additional Devices/ Invasiveness | Dependence on Motor Specifications | Signal Sampling Rate | Training Data Requirement |
|---|---|---|---|---|---|---|---|---|
| **PM** | Sensor-based | | Low | Very High 1.5 °C [6] | Yes/Mechanical modification | Low | Low | No |
| | Model-based | Thermal models | High | High 3.3 °C [19] | No | High | Medium | No |
| | | Electrical models | Medium | High 2 °C [87] | No/Signal injection | Medium | High | No |
| | | Mechanical models | Medium | Medium 4 °C [35] | No/Signal injection | Medium | High | No |
| | AI algorithm-based | | High | High 1.5 °C [37] | No | Low | Medium | Yes |
| **Stator Winding** | Sensor-based | | Low | Very High 2 °C [7] | Yes/Mechanical modification | Low | Low | No |
| | Model-based | Thermal models | High | High 2.3 °C [52] | No | High | Medium | No |
| | | Electrical models | Medium | Medium 4 °C [86] | No/signal injection | Medium | High | No |
| | AI algorithm-based | | High | Medium 4.5 °C [37] | No | Low | Medium | Yes |

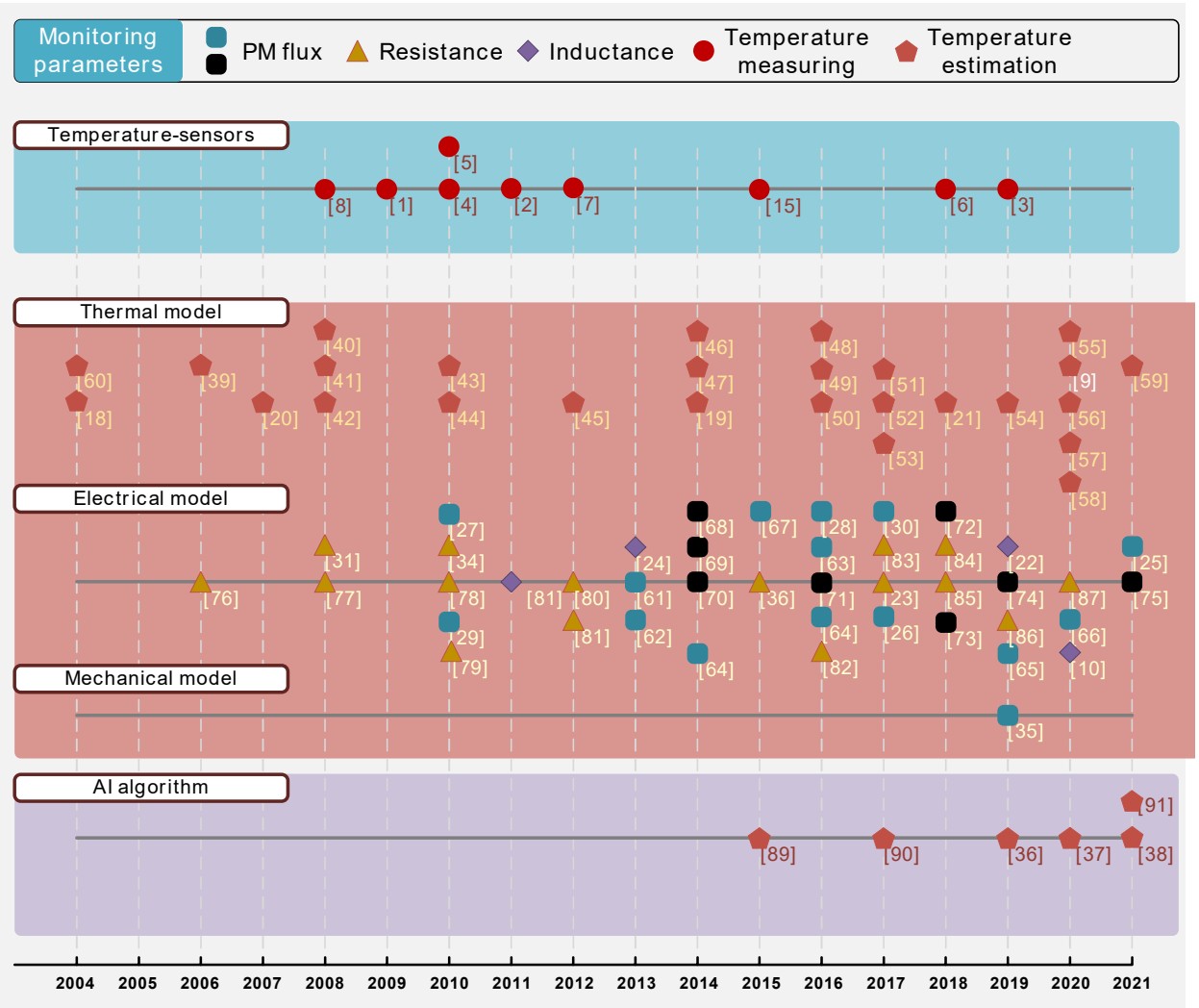

**Figure 4.** Development of the temperature monitoring methods for PM machines [1–10,15,18–31,34–87,89–91].

**Author Contributions:** Conceptualization, P.Z. and T.M.; methodology, P.Z.; software, T.M.; validation, T.M., P.Z.; formal analysis, P.Z.; investigation, T.M.; resources, T.M.; data curation, T.M.; writing—original draft preparation, T.M.; writing—review and editing, P.Z.; project administration, P.Z.; funding acquisition, P.Z. All authors have read and agreed to the published version of the manuscript.

**Funding:** The research was funded by National Key Research and Development Program of China (2018YFB130089), and the National Science Foundation under Grant 51822705 and 51777112.

**Institutional Review Board Statement:** Not applicable.

**Informed Consent Statement:** Not applicable.

**Data Availability Statement:** Not applicable.

**Conflicts of Interest:** The authors declare no conflict of interest.

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
