# Peer review of "A Review of Thermal Monitoring Techniques for Radial Permanent Magnet Machines"

_machines, doi:10.3390/machines10010018_

Round 1
Reviewer 1 Report
this paper presents A Review of Thermal Monitoring techniques for Permanent Magnet Machines, it is a review, . this topic is very interesting , it is a hard work, but it need some revison
1)References are not standard, why has the introduction not cited references? why dosed the cited reference begin from [27]. the nummber of references is confusing in the literature
2) The literature needs to be enriched, because the topic Thermal Monitoring techniques for Permanent Magnet Machines is a long-term topic,it need a systematically analyse for its last ,now ,future, many efficient method ,arising some new problem.
Reviewer 2 Report
The paper is well-written and organized. A qualitative overview of thermal monitoring techniques for PM machines is provided. In my opinion, the paper should be accepted in its present form.
Reviewer 3 Report
Title: A Review of Thermal Monitoring techniques for Permanent Magnet Machines
Summary:
This paper fully and comprehensively summarizes the research status of thermal monitoring about permanent magnet and stator winding of permanent magnet machines. Meanwhile, the outlook for the future is also discussed.
Comments:
- In the introduction, the abbreviation “PM” appears for the first time and should be explained.
- Various schemes of thermal monitoring in the introduction are introduced, but few references are cited, which cannot support your discussion. Other sections have similar problems.
- Section 3 mainly focuses on sensor‐based methods of temperature monitoring for PM. You should introduce several typical methods and their connections and differences. Ref. [32], [103], [4], and [57] do not address these issues.
- In Eq. (3), the parameters ,r,φ, z, should be explained.
- When describing the model-based method in Fig. 3, it is recommended that the introduction of each method be divided into PM temperature monitoring methods and stator windings temperature monitoring methods.
- Section 4.1 introduces the thermal model of PM machines. Why is the title only about the thermal model of PM? PM machines consists of PM and stator windings in this paper. Please carefully check whether there are the same chaotic expressions in other places.
- In each sectional discussion, you should cite some references, discuss the views of other scholars in this field, and put forward your own views.
- More AI algorithm‐based methods should be analyzed. You need to introduce in detail the input and output parameters of AI algorithms in various references and the solution effect.
- Some references in recent years are missing, such as “Y. S. Li, C. B. Li, A. Garg, L. Gao, and W. Li, Heat dissipation analysis and multi-objective optimization of a permanent magnet synchronous motor using surrogate assisted method, Case Studies in Thermal Engineering, vol. 27, Oct, 2021.”
Reviewer 4 Report
The paper is well organized and offers an exhaustive review of the thermal monitoring techniques. However, some parts seem too general and not supported of a more insight about the main issues, especially concerning experimental approaches. I think that a reader would like to have more practical details about the effectiveness of one technique with respect to the others. I suggest inserting more significant results taken from the cited literature which is complete and well organized. Other points that should be considered are:
- It is quite difficult to follow not ordered citations, then it is preferable to order references according to citation.
- Most of the presented techniques seem more focused on radial flux machines (see for instance equation (4)); it must be advisable to discuss the applicability about axial flux machines which are more and more considered for many applications; alternatively, the authors can consider changing the title mentioning only radial flux permanent magnet machines.
- Discussion on the losses in paragraph 2.1 should be improved according to the following remarks:
- Equation (1) applies only with alternating component, but PMs experiment also a DC component; therefore, Bm cannot be the maximum one; furthermore (1) is quite similar to the one generally used for laminations, then it applies well only with very thin PMs; please revise description providing a more insight literature about PM losses
- PMs heating also comes from the losses of the rotor magnetic circuit; then, (2) should be anticipated;
- Equation (2) needs to be explained as general formulation of the iron losses is much more complicated; alternatively, the authors can insert some references;
- a note could be included about the role of windage losses, very important especially in high-speed machines
- Section 3 discussion should be enriched showing some practical implementation of the sensing equipment.
- In Section 4, it is not clear how preliminary FEM analyses (FEAs) can be used for an online temperature monitoring. FEAs can predict temperature distribution by combined CFD and electromagnetic solutions; however, it is more difficult to use it for monitoring purposes. Please provide more details on this issue
- There are a several symbols used for PM thermal coefficients (alpha_m, alpha_B, alpha_T, alpha_PM, beta). Please clarify the meaning for each one if they are different, otherwise select only one for all the discussion
- From line 39 to line 54, some concepts seem repeated; I suggest revising to achieve a more organized presentation
- The sentence from line 182 to 184 is not clear
- I prefer comma after equations to be on the same line; the equations should be punctuated as regular text according to the template format.
Finally, there are some typos:
- Line 139: “wight” -> “weight”
- Line 147: “. the” -> “. The”
- Line 179: “disscussion” -> “discussion”
- Line 241: “indictor” -> “indicator”
- Line 303: “described” -> “describe”
- Line 460 and 698: “expect” -> “except”
- Line 534: “nonlinear” -> “nonlinearity”
- Line 550: “the speed” -> “that the speed”
- Line 680: “limit” -> “limiting”
Reviewer 5 Report
COMMENT TO THE AUTHORS
General comments
All the references should be cited in the paper in order.
The references should be ordered according to the proposed technique. The main novelty of the references should be included in your paper.
Some parts of the paper are too generic.
Additionals commnets
1.-Line 139 Please confirm the word wight
Wight ?
2.-Line 147 Capital letter.
self‐temperature. the practical influence
3.-Line 165 Do you mean “To perform” instead “To realize”?
4.-Fig. 2 is not related to thermal monitoring. It would be much better to include a figure related to the thermal monitoring.
5.-Line 219 Please clarify
The red dots represent the temperature sensors.
6.-Section 3
This section is too ambiguous. It should be more precise including more information for example about the position of the sensors and temperatures measured.
It would be useful to include the position of the sensors and the temperature measured, for example stator windings, iron core, bearings, cold air and hot air... etc.
It would be nice to have some data about real machines, what can you find in the market?
7.-Equation 5
Does the stator resistance depend on the temperature?
8.-Thermal models
In section 4.1 the thermal models are explained. However, it is not clear how the machine losses are taken into consideration. Equation 5 takes into consideration stator losses, but what about the iron losses and the losses in the magnets?
9.-Equation 9
Have you obtained this equation from any reference?
10.-Estimation temperatures based on the high frequency resistance measurement.
It not clear, witch parameters should be measured and calculated?
For example, I understand you should measure the DC resistances of the winding at a reference temperature.
11.-Equation 14
Is this equation only appropriate for DC? If yes, it would be nice to clarify.
12.-Line 524 The section should be 4.2.3
5.2.3 Sectional discussion
13.-Lines 593- 595 AI input data for learning.
In case an AI device should estimate the temperatures of a machine in different operation conditions, the required data are, the temperatures and the operation conditions (electrical, mechanical or both)
I don’t understand how it is possible to estimate the temperature only measuring the electrical or only the mechanical quantities.
On the other hand, if only thermal quantities are measured, it is not possible to identify the operation conditions.
Please clarify this sentence.
In general, the AI algorithm‐based methods are based on collecting extensive measurements of quantities and target parameters, which is independent of motor specifications and mathematical models. These quantities can be electrical, mechanical or thermal parameters.
Round 2
Reviewer 4 Report
The authors provided adequate responses to the issues raised in the previuos submission.